# Glycated Albumin and Continuous Glucose Monitoring Metrics in Dogs with Diabetes Mellitus: A Pilot Study

**DOI:** 10.3390/ani15142004

**Published:** 2025-07-08

**Authors:** Soon-Chan Kwon, Ju-Hyun An, Dong-Hoo Kim, Hwa-Young Youn

**Affiliations:** 1Laboratory of Veterinary Internal Medicine, Department of Veterinary Clinical Science, College of Veterinary Medicine, Seoul National University, Seoul-si 08826, Republic of Korea; ra645@naver.com; 2Kwon & Jung Suwon Animal Medical Center, Suwon-si 16698, Republic of Korea; 3Department of Veterinary Emergency and Critical Care Medicine, College of Veterinary Medicine, Kangwon National University, Chuncheon-si 24341, Republic of Korea; moonlit0816@naver.com; 4Gogang Animal Hospital, Bucheon-si 14408, Republic of Korea; gogangah@hanmail.net

**Keywords:** diabetic mellitus, dog, continuous glucose monitoring system, glycated albumin

## Abstract

Diabetes mellitus in dogs occurs when the animal does not produce enough insulin, causing high blood glucose levels, which can damage organs. Common tests like hemoglobin A1c (HbA1c) can provide information on longer-term blood glucose levels but may overlook short-term changes. We explored glycated albumin (GA), which reflects average blood glucose over about two to three weeks. In our study, we compared GA, fructosamine, HbA1c, and continuous glucose monitoring (CGM) data in both diabetic and healthy dogs. We found that GA closely correlated with the fructosamine and HbA1c tests and tracked well with CGM results. This suggests that GA could help veterinarians to better understand and manage short-term blood glucose control.

## 1. Introduction

According to the ALIVE consensus definitions, canine diabetes mellitus is classified etiologically as either insulin-deficient (IDDM) or insulin-resistant (IRDM), with IDDM remaining the most common form seen in practice [1]. A deficit in insulin levels results in elevated blood glucose levels, which can lead to complications affecting the eyes, kidneys, nerves, heart, and blood vessels. Since there is currently no cure for IDDM, the blood glucose levels of canine patients should be monitored using a glucometer, and glucose indicators, such as fructosamine and hemoglobin A1c (HbA1c), have to be used to determine appropriate insulin dosages [2]. However, recent clinical audits report that almost one-quarter of insulin-treated dogs are still struggling to achieve target glycemic control, emphasizing a practical monitoring gap that standard tools fail to close.

Despite these standard monitoring tools, managing canine diabetes can be challenging. Single-time glucose measurements may not accurately reflect daily fluctuations or long-term trends in glycemic control. Fructosamine is useful for assessing average glucose over about two weeks, while HbA1c offers insights into glucose regulation over roughly two to three months [3]. However, both tests have practical limitations: fructosamine can be affected by changes in serum protein levels, particularly in hypoalbuminemia or protein-losing disorders, and HbA1c may not capture short-term variability in glucose [4]. Thus, there is a growing need for tools that capture continuous or near-continuous changes in glucose and provide a more complete picture of a patient’s metabolic status.

Continuous glucose monitoring systems (CGMSs) allow the real-time monitoring of glucose levels, enabling timely responses to glucose fluctuations [5]. These small sensors, placed beneath the skin, measure interstitial fluid glucose and help to refine insulin dosages or dietary plans by generating critical alerts for significant deviations [6]. CGMSs can also aid in identifying asymptomatic hypoglycemia or hyperglycemia, ultimately enhancing insulin therapy for each dog [7]. However, the cost and technical demands of CGMSs may limit their routine application in many veterinary settings, prompting a search for additional short-term biomarkers that are simpler to measure.

Recently, glycated albumin (GA) was proposed as a novel biomarker for diabetes in dogs. In human medicine, GA has often been used alongside fructosamine, especially in conditions where HbA1c measurements are unreliable (e.g., pregnancy, renal failure, and anemia) [8]. GA reflects short-term changes in blood glucose levels, corresponding to the 2–3-week half-life of albumin, and research in dogs suggests that GA levels correlate with fructosamine [9]. Moreover, GA is reported to be minimally affected by transient stress-induced hyperglycemia or daily glucose fluctuations, making it a reliable indicator of glycemic control over 1–3 weeks [10]. Comparable short-term serum assays have already improved the recognition of endocrine-related dysglycemia in diabetic cats, highlighting the clinical value of inexpensive markers [11].

We hypothesized that GA would align closely with CGM-derived measures, thereby serving as a valuable adjunct to existing biochemical markers for evaluating and optimizing glycemic control in canine diabetes. Considering the need for additional biomarkers that capture short-term glucose variability, integrating CGM metrics with GA levels may offer new insights into the day-to-day management of diabetic dogs. The objectives of this study were to identify and compare GA and continuous glucose monitoring (CGM) metrics in diabetic dogs and control groups and to evaluate the association between GA levels and CGM metrics in diabetic dogs.

Because this investigation was intentionally designed as a pilot study with a relatively small sample size, all findings should be regarded as preliminary and will need confirmation in larger, independent canine cohorts.

## 2. Materials and Methods

### 2.1. Study Population

This prospective study was conducted between October 2023 and December 2024 at two veterinary hospitals in Suwon-si and Bucheon-si, Republic of Korea. The experiment involved canine patients whose owners consented to participate. Additionally, a trial was conducted to compare CGM metrics with other blood glucose indicators in dogs whose owners consented to CGMS use, provided they met specific criteria. For dogs with IDDM, the inclusion criteria were a confirmed diagnosis of IDDM at a veterinary hospital, management of diabetes with exogenous insulin and diet, and no change in insulin regimen for at least 1 month. The exclusion criteria included dogs hospitalized for acute diabetes complications, such as diabetic ketoacidosis (DKA) and hyperosmolar hyperglycemic state (HHS), dogs hospitalized for other diseases, and those whose insulin dosage was significantly altered due to comorbidities during the study period. Dogs participating in the additional study returned to the hospital 14 days after CGMS administration, where blood glucose-related indicators were measured using blood tests for comparison.

### 2.2. Measurements

The clinical assessment of the patients was based on a scoring system previously described by [12]. Questions regarding clinical signs, presence of complications, and treatment compliance were scored on a scale of up to 9 points, while the veterinarian’s physical examination contributed up to 3 points, resulting in a total possible score of 12 (Appendix A). Blood tests were performed immediately after sample collection. Complete blood count (CBC) and HbA1c tests were conducted using whole blood collected in EDTA tubes. Plasma samples for blood urea nitrogen (BUN), creatinine, albumin, total protein, GA, and fructosamine tests were obtained by collecting blood in heparin tubes followed by centrifugation. CBC tests were conducted using IDEXX’s Procyte Dx analyzer. BUN, creatinine, albumin, and total protein levels were measured using the Fuji NX700 analyzer (FUJIFILM Corporation, Tokyo, Japan). Fructosamine concentration was determined using the IDEXX Catalyst Dx analyzer (IDEXX Laboratories, Inc., Westbrook, ME, USA). GA and HbA1c tests were conducted using a CareSign-V Analyzer (i-Sens, Seoul, Republic of Korea) with a CareSign-V GA and HbA1c cartridge. GA levels were measured using a cartridge-based system specifically validated for use in dogs and cats. The assay required 5 μL of serum or plasma treated with lithium heparin.

### 2.3. CGM System

Eleven dogs were fitted with CGMS devices to monitor mean blood glucose levels, glucose ranges, and overall glucose trends. CGMS data traces < 10 days were discarded; dogs were refitted with a new sensor and included only if the replacement yielded ≥ 10 uninterrupted days of CGM data, during which GA, fructosamine, and HbA1c were sampled. Two continuous glucose monitoring (CGM) systems originally developed for human use were employed in this study: the FreeStyle Libre 2 (Abbott Diabetes Care, Alameda, CA, USA; n = 1), which has previously been tested in both healthy and diabetic dogs [6,7,13], and the CareSens Air (i-SENS, Seoul, Republic of Korea; n = 10), a 15-day sensor introduced in 2023 that utilizes the same interstitial fluid–based technology, which has so far only been evaluated in humans [14] and was applied to dogs for the first time in this study. CGM outcomes were quantified using the following metrics: mean glucose levels (expressed in mg/dL) and the percentage of time spent in various glucose ranges. In the absence of canine-specific CGM guidelines, we selected two practical bands, 90–130 mg/dL (fasting normoglycemia) and 90–250 mg/dL (routine clinical target range for insulin-treated dogs). The ranges included time in range (TIR1) for glucose levels between 90 and 130 mg/dL, TIR2 for levels between 90 and 250 mg/dL, time above range (TAR1) for levels exceeding 130 mg/dL, TAR2 for levels exceeding 250 mg/dL, and time below range (TBR) for levels below 90 mg/dL.

### 2.4. Ethical Approval

This study was conducted in accordance with protocols approved by the Institutional Animal Care and Use Committee of Kangwon National University (KNU), Republic of Korea, and in compliance with authorized guidelines (protocol no. KW-231027-2).

### 2.5. Statistical Analyses

All statistical analyses were conducted in collaboration with the Statistical Research Institute at Seoul National University (Seoul, Republic of Korea). Prior to formal analysis, the distribution of each continuous variable was assessed using the Shapiro–Wilk test. Variables with *p*-values ≥ 0.05 were considered normally distributed, and those with *p* < 0.05 were treated as non-normally distributed. For normally distributed variables, a parametric method such as Student’s *t*-test was used. For non-normally distributed data, non-parametric tests (e.g., Mann–Whitney U test) or resampling methods (e.g., permutation tests) were applied. To compare the distribution of each variable between the diabetic and control groups, resampling methods (e.g., permutation tests) were employed without assuming normality or homogeneity of variance, particularly in consideration of the small sample size.

Correlations between glycemic biomarkers (GA, fructosamine, HbA1c) and CGM-derived metrics (mean glucose, TIR, TAR, TBR) were analyzed using Pearson’s correlation when both variables met the normality assumption, and Spearman’s rank correlation otherwise.

ROC curve analysis was performed to evaluate the discriminative ability of glycemic markers, with the area under the curve (AUC) used as a summary measure. Optimal thresholds were identified using the Youden index and the minimum distance method.

All statistical analyses were conducted using R (version 4.3.2; R Foundation for Statistical Computing, Vienna, Austria; https://cran.r-project.org/, accessed on 2 April 2025) and GraphPad Prism version 7 (GraphPad Software, Inc., San Diego, CA, USA). *p*-value < 0.05 was considered statistically significant.

## 3. Results

### 3.1. Patient Characteristics

Thirty dogs participated in this study and were divided into two groups: an experimental group of 10 diabetic dogs and a control group of 20 healthy dogs. Information regarding the participating dogs is summarized in Table 1 and Appendix A.

### 3.2. Correlation Between GA and Fructosamine and HbA1c

Correlations were evaluated using blood glucose-related markers in the 30 dogs participating in this study. Linear regression analyses illustrating the relationships between GA levels and the two key glycemic biomarkers demonstrated a significant correlation between GA and fructosamine (*r* = 0.7040, *p* < 0.0001) and between GA and HbA1c (*r* = 0.7249, *p* < 0.0001) (Figure 1).

### 3.3. Correlation Between Glycemic Markers and CGM Metrics in Dogs

Among the dogs participating in the trial, the correlation between CGM metrics and glycemic markers was evaluated in those whose owners consented to CGMS fitting (Figure 2). Eleven dogs participated in these additional experiments, including seven diabetic and four control dogs. Blood tests were performed on dogs fitted with CGMS to rule out potential factors that could affect the results, confirming the absence of anemia, hypoalbuminemia, or azotemia in either group (Table 2).

The GA levels, fructosamine levels, HbA1c percentages, and CGM results are summarized in Table 3. In diabetic dogs, the median GA level on day 14 was 29.10%, while in the control group, it was 17.10%. Consistent and statistically significant between-group differences were also documented for fructosamine and HbA1c. The median fructosamine level on day 14 was 303.00 μmol/L in the diabetic group, compared to 214.00 μmol/L in the control group. The median HbA1c levels were recorded at 4.2% in diabetic dogs and 2.1% in control dogs. All CGM metrics showed significant differences between diabetic and control groups, including mean glucose levels, TIR1, TIR2, TAR1, TAR2, and TBR, suggesting distinct glycemic patterns in the two groups.

The correlations among GA, fructosamine, HbA1c, and various CGM metrics are presented in Figure 3, Figure 4 and Figure 5. To evaluate the association between glycemic biomarkers and CGM-derived metrics, normality was first assessed for all variables using the Shapiro–Wilk test. GA, fructosamine, HbA1c, mean glucose, TIR2, and TAR2 were found to be normally distributed (*p* ≥ 0.05), whereas TIR1, TAR1, and TBR did not meet the normality assumption (*p* < 0.05). Pearson’s correlation coefficient was used when both variables in a pair satisfied the assumption of normality; otherwise, Spearman’s rank correlation coefficient was applied. For GA, statistically significant associations were confined to mean glucose (*r* = 0.6924, *p* = 0.0182), TIR2 *(r* = –0.6026, *p* = 0.0498) and TAR2 (*r* = 0.7058, *p* = 0.0152); the links with TIR1, TAR1 and TBR did not reach the 0.05 threshold. Fructosamine levels were significantly correlated with mean glucose (*r* = 0.6257, *p* = 0.0395), TIR1 (*r* = –0.7182, *p* = 0.0168), TAR1 (*r* = 0.8055, *p* = 0.0028), TAR2 (*r* = 0.6386, *p* = 0.0344) and TBR (*r* = –0.6287, *p* = 0.0383); their association with TIR2 (*r* = –0.4135, *p* = 0.2062) was not significant. HbA1c demonstrated strong correlations with mean glucose (*r* = 0.9043, *p* = 0.0001), TIR1 (*r* = −0.7730, *p* = 0.0053), TIR2 (*r* = −0.7606, *p* = 0.0066), TAR1 (*r* = 0.7924, *p* = 0.0036), TAR2 (*r* = 0.9035, *p* = 0.0001) and TBR (*r* = –0.6758, *p* = 0.0225).

In Figure 3, Figure 4 and Figure 5, scatter plots show the correlations between each glycemic marker. Correlation coefficients (r) and *p*-values are shown for each comparison. Pearson’s correlation coefficient was used when both variables were normally distributed according to the Shapiro–Wilk test; otherwise, Spearman’s rank correlation coefficient was applied. Solid lines represent the fitted linear regression, and shaded areas (if present) indicate the 95% confidence interval. Statistical significance was defined as *p* < 0.05. Abbreviations: HbA1c, glycated hemoglobin A1c; TAR1, time above range > 130 mg/dL; TAR2, time above range > 250 mg/dL; TBR, time below range < 90 mg/dL; TIR1, time in range 90–130 mg/dL; TIR2, time in range 90–250 mg/dL.

### 3.4. ROC Curve Analysis

To further evaluate the diagnostic performance of GA in distinguishing diabetic from control dogs, a receiver operating characteristic (ROC) curve analysis was performed (Figure 6). The area under the curve (AUC) was 0.95 (95% CI: 0.91–1.00, *p* < 0.0001), indicating excellent diagnostic accuracy. Based on the Youden Index, the optimal GA cut-off value was 17.4%, which provided a sensitivity of 88.2% and a specificity of 90.0%. These findings suggest that GA can reliably differentiate diabetics from control dogs and may be useful for clinical decision-making regarding glycemic control. For completeness, the minimum-distance method identified an alternative cut-off of 21.8%, which produced a slightly higher overall accuracy at the expense of marginally lower sensitivity.

## 4. Discussion

This study aimed to evaluate the monitoring utility of GA by comparing it with the established biomarkers fructosamine and HbA1c and by analyzing their correlations with CGM metrics. We also assessed these markers against CGM metrics, which are new and useful indicators for diagnosing and managing diabetes.

In human medicine, traditional diabetes management has primarily relied on blood glucose measurements following fasting or the oral glucose tolerance test (OGTT). However, these methods have been criticized for capturing only transient glycemic states and showing limited reproducibility in both individual assessments and population-wide studies [15]. Over four decades ago, glycosylated hemoglobin (HbA1c) became a pivotal biomarker for estimating average blood glucose levels, eventually establishing itself as a benchmark for diabetes monitoring. Despite its widespread use, HbA1c has certain limitations; it does not identify the causes of elevated glucose levels and cannot accurately reflect short-term or rapid metabolic changes. Consequently, alternative biomarkers, such as fructosamine and GA, have been proposed to address these gaps [8].

GA reflects glycemic exposure over a period of 2–3 weeks, aligning with the half-life of albumin. Several canine studies have characterized GA [9,10]. Comparable work in humans has likewise demonstrated strong concordance between GA and CGM metrics over a 2–3-week interval [16], and feline data indicate that GA correlates well with short-term glycemic control and can complement fructosamine in diabetic cats [17]. These cross-species data support GA as a broadly applicable intermediate-term marker. GA is also reported to remain stable despite diurnal glucose swings in both dogs and humans. However, no prior research has examined the correlation between CGMS and glycated albumin in dogs, leaving a gap in understanding how these metrics might collectively inform clinical practice.

Although fructosamine and GA both reflect short-term glycemic status (~2–3 weeks), fructosamine may be more influenced by total serum protein changes, potentially leading to the under- or overestimation of glycemia in dogs with hypoproteinemia or fluctuating protein levels [18]. By contrast, GA specifically measures albumin glycation as a proportion of total albumin, making it less susceptible to confounding due to altered total protein [19]. HbA1c remains valuable for evaluating longer-term (~2–3 months) glycemic trends but may not capture rapid fluctuations or short-term treatment adjustments. Moreover, conditions that affect red blood cell turnover or hemoglobin metabolism can invalidate HbA1c results [4]. Therefore, GA may offer advantages in clinical scenarios where serum protein or RBC-related factors hinder the reliability of other markers.

Because TIR, TAR, and TBR are mutually dependent time-percentage metrics, even slight threshold changes can flip the sign or strength of their correlations; therefore, results must be interpreted with caution. GA showed significant association only with mean glucose, TIR2 and TAR2, underscoring its utility as a short-term marker of overall hyperglycemic exposure when full CGM data are unavailable. The lack of association with TIR1 or TAR1 is expected because these metrics include euglycemic readings that do not influence albumin glycation to the same extent.

In this study, we investigated the relationships among GA, fructosamine, and HbA1c levels, confirming a significant correlation between GA and fructosamine, as well as between GA and HbA1c. Additionally, correlations between CGM metrics and blood glucose-related markers were analyzed in dogs fitted with CGMS devices. The analysis of mean blood glucose and glycemic range metrics via CGM indicated significant differences between control and diabetic dogs. Specifically, mean blood glucose level, TIR1, TAR1, and TAR2 were extremely significant (*p* < 0.001); TIR2 was highly significant (*p* < 0.01); and TBR was significant (*p* < 0.05), indicating robust associations. These results highlight the value of CGM in providing a more comprehensive picture of day-to-day glycemic control compared to single-time-point blood glucose measurements [20].

After analyzing the correlation between GA and various CGM metrics in our study, we found that GA was significantly associated with mean glucose level (r = 0.6924, *p* = 0.0182), TIR2 (r = –0.6026, *p* = 0.0498), and TAR2 (r = 0.7058, *p* = 0.0152), but not with TIR1, TAR1, or TBR. Interestingly, a 2020 study [7] proposed two main criteria for good glycemic control in dogs: (1) achieving at least 50% of readings within a specified target range and (2) having ≥60% of readings in the target range with fewer than 30% above that range. Although these criteria were described in a slightly different context, the principle behind the second criterion—quantifying the proportion of time spent within and above ideal glucose levels—parallels our use of time in range 2 (TIR2) and time above range 2 (TAR2). In our current study, both TIR and TAR showed notable correlations with GA, suggesting that GA could be a helpful indicator for short-term glycemic control when combined with CGM data.

One limitation of GA is its dependence on stable albumin turnover. In conditions such as protein-losing nephropathy or hepatic insufficiency, albumin half-life may be significantly altered, potentially causing GA values to under- or overestimate actual glycemia. While fructosamine is likewise affected by serum protein levels, the specific nature of GA (i.e., a ratio of glycated albumin to total albumin) can sometimes minimize confounding—yet severe loss or reduced synthesis of albumin can still skew results. Therefore, in dogs with concurrent liver disease or severe protein-losing enteropathy, GA should be interpreted cautiously alongside other markers such as fructosamine or HbA1c.

In feline medicine, the use of a simple IGF-1 ELISA has tripled the detection rate of hypersomatotropism, demonstrating how an affordable serum test can significantly improve case identification [11,21]. Similarly, GA could serve as a rapid screening tool in general practice, with CGM reserved for complex or referral cases.

Nevertheless, our study had several limitations that need to be addressed in future research. We chose 130 mg/dL as a cutoff for TIR somewhat arbitrarily. Establishing evidence-based cutoffs would allow for improved standardization across studies and clinical settings [7]. A larger-scale investigation using more dogs and a formal ROC analysis would strengthen the reliability of any proposed targets or correlations, thereby clarifying how GA can be best utilized in routine clinical settings.

Second, our sample size was relatively small, and we did not include dogs with multiple comorbidities or conditions that might affect albumin turnover or red blood cell lifespan. These factors could influence GA, fructosamine, or HbA1c levels and potentially alter their correlations with CGM metrics [8]. Including a broader population of canine patients would help to validate the clinical applicability of GA and other glycemic indices in different scenarios, such as kidney disease, protein-losing nephropathy, or systemic inflammatory conditions. Therefore, further studies with larger cohorts, various comorbidities, and broader clinical contexts are required to validate the clinical applicability of GA and establish evidence-based guidelines for its integration into routine diabetes management protocols in canine practice. In addition, GA-based correction formulas that have been proposed in human medicine should be evaluated or adapted for use in dogs [22].

Furthermore, although our pilot ROC analysis suggested a cut-off value of approximately 17.4% for GA (AUC = 0.95, 95% CI: 0.91–1.00, *p* < 0.0001), its clinical utility should be interpreted cautiously due to the limitations inherent in our study design. Our diabetic group consisted of dogs already diagnosed and relatively well-controlled for at least one month, whereas the control group was clearly non-diabetic, resulting in two polarized populations. Thus, our results do not fully represent the diverse presentations encountered in clinical practice, such as borderline hyperglycemia or newly suspected diabetes. Previous studies have reported various GA cut-off values ranging approximately from 13.4% to 29.2% depending on the purpose, such as distinguishing diabetic from non-diabetic dogs or categorizing different levels of glycemic control [9,10]. Therefore, although our cut-off is within this previously reported range, further large-scale studies, including dogs with uncertain glycemic status, are required to confirm whether our threshold reliably differentiates diabetic from non-diabetic cases at initial presentation.

Lastly, although CGM offers a more detailed look at real-time glucose fluctuations, its cost and the need for technical expertise can limit widespread adoption in veterinary practice [23]. The integration of GA measurement, which can be more straightforward and less time-intensive, might serve as a practical bridge between traditional spot checks and full-scale CGM, especially when owners cannot commit to round-the-clock monitoring. By corroborating short-term trends with GA and capturing diurnal variations with CGM, veterinarians could develop more refined insulin dosing regimens and better detect asymptomatic hypoglycemia or hyperglycemia in diabetic dogs.

## 5. Conclusions

In conclusion, despite the limitations of our pilot study, including a small sample size and a relatively homogeneous population, our findings suggest that glycated albumin (GA), alongside fructosamine and HbA1c, offers valuable insight into glycemic control over a two-week period. Notably, GA showed strong associations with both mean glucose level and CGM-derived extremes such as TIR2 and TAR2, supporting its potential as a complementary short-term glycemic marker in diabetic dogs. Given GA’s relative stability in conditions affecting serum protein or red blood cell turnover, it may offer clinical advantages in patient populations where traditional glycemic markers are less reliable or practical.

## Figures and Tables

**Figure 1 animals-15-02004-f001:**
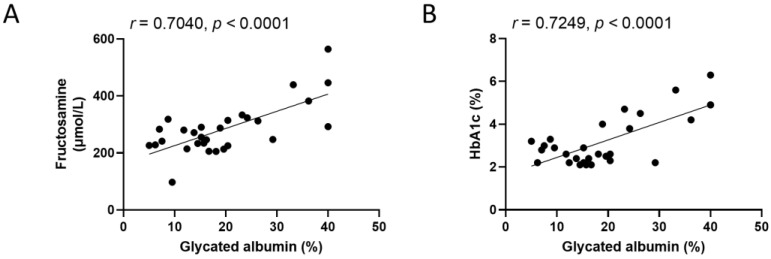
Correlation of glycated albumin with fructosamine and HbA1c in dogs enrolled in this study (n = 30). (**A**) Scatter plot showing the correlation between glycated albumin and fructosamine. (**B**) Scatter plot showing the correlation between glycated albumin and HbA1c. Both variable pairs satisfied the assumption of normality based on the Shapiro–Wilk test and were analyzed using Pearson’s correlation coefficient. Solid lines represent linear regression fits; *r* and *p*-values are indicated for each comparison. A statistically significant correlation was defined as *p* < 0.05.

**Figure 2 animals-15-02004-f002:**
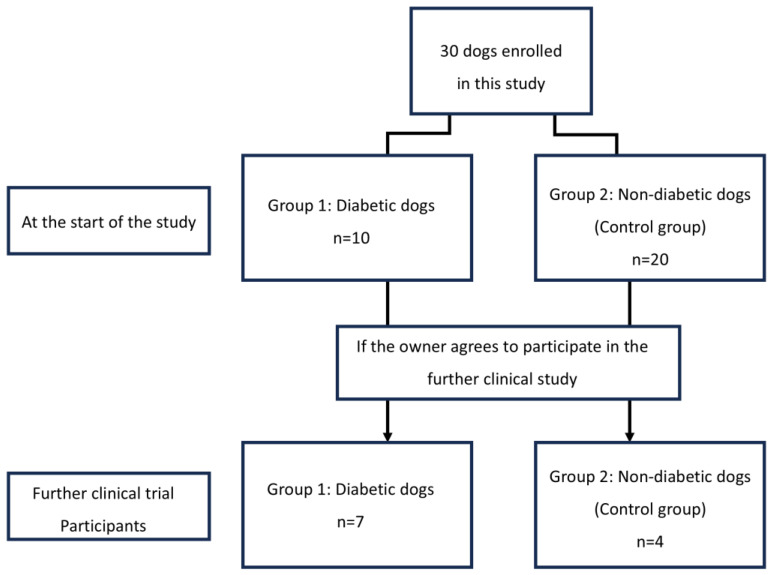
Flow diagram of changes in the group composition of dogs enrolled in this study. In this study, 30 dogs were divided into two groups according to diabetic and non-diabetic.

**Figure 3 animals-15-02004-f003:**
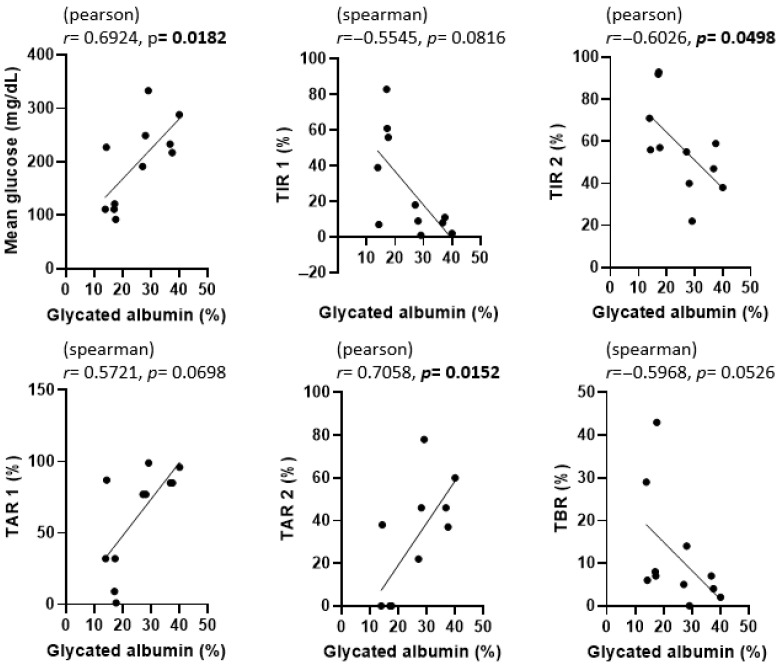
Correlation between glycated albumin (GA) and CGM metrics in dogs enrolled in this study. *p*-values less than 0.05 were considered statistically significant and are indicated in bold.

**Figure 4 animals-15-02004-f004:**
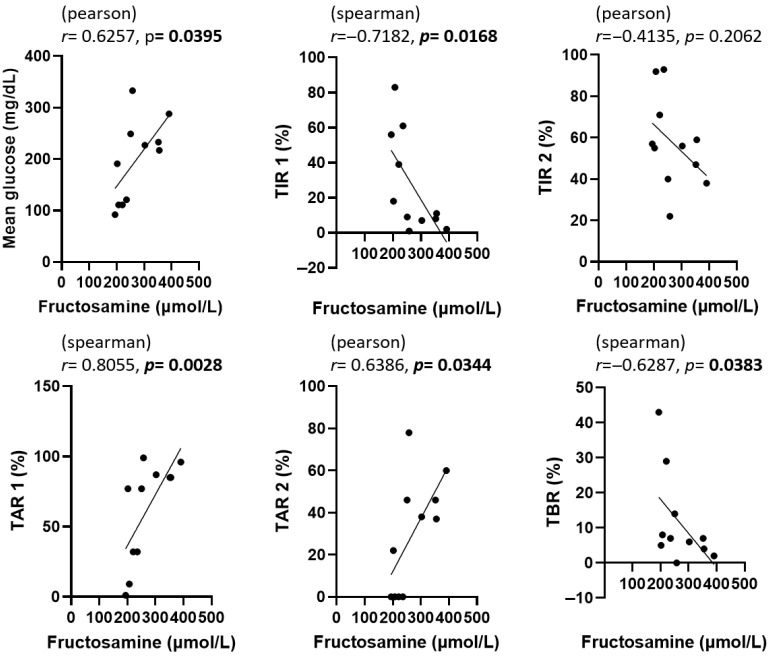
Correlation between fructosamine and CGM metrics in dogs enrolled in this study. The statistical methods and abbreviations used are identical to those in Figure 3. *p*-values less than 0.05 were considered statistically significant and are indicated in bold.

**Figure 5 animals-15-02004-f005:**
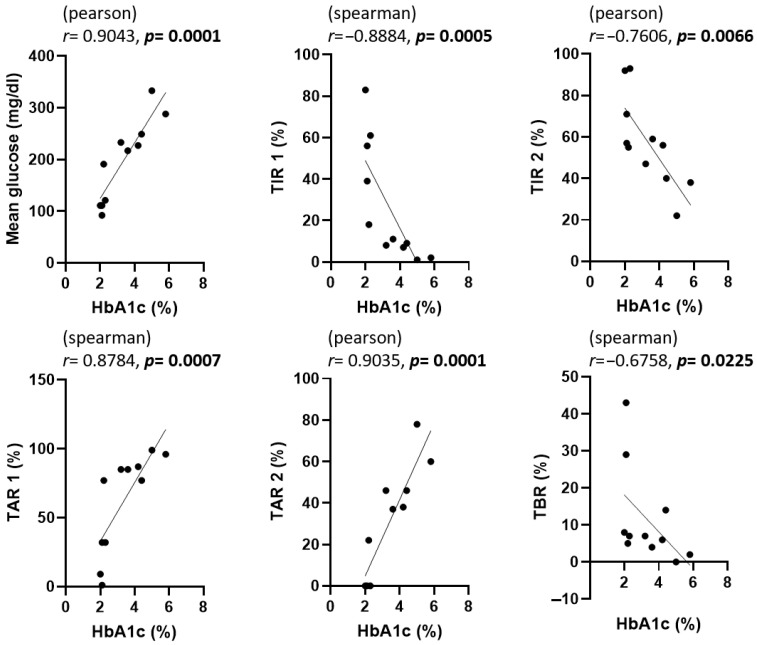
Correlation between HbA1c and CGM metrics in dogs enrolled in this study. The statistical methods and abbreviations used are identical to those in Figure 3. *p*-values less than 0.05 were considered statistically significant and are indicated in bold.

**Figure 6 animals-15-02004-f006:**
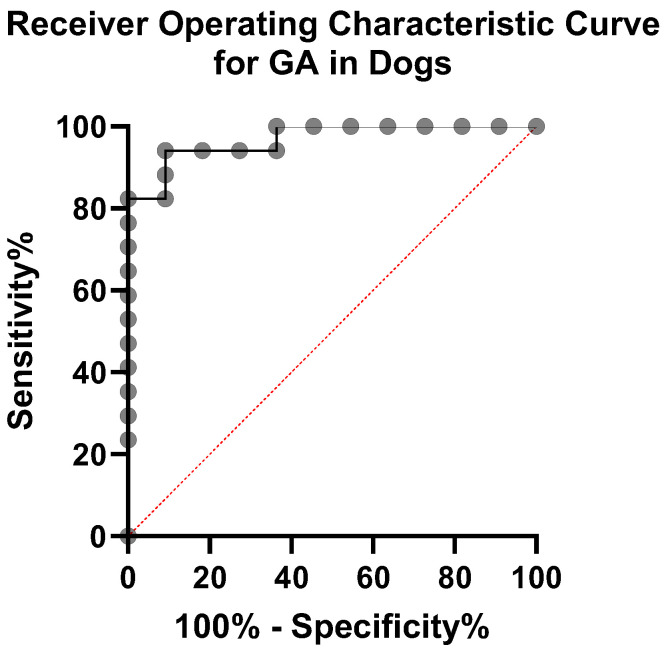
ROC curve for GA. The diagonal dashed line represents chance performance (AUC = 0.5).

**Table 1 animals-15-02004-t001:** Signalment of dogs with diabetes mellitus and non-diabetic dogs.

Variables	Groups	
	Diabetic Mellitus (n = 10)	Control (n = 20)
**Breed**	Bichon (2), Golden Retriever (1), Jindo (1), Maltese (3), Mixed (1), Pomeranian (1), Poodle (1)	Bichon (1), Chihuahua (1), Dachshund (1), French bulldog (1), Maltese (3), Mixed (2), Pomeranian (2), Poodle (6), Schnauzer (1), Shih tzu (1), Spitz (1)
**Sex**	CM (2), F (2), M (1), SF (5)	CM (6), F (2), M (1), SF (10), Unknown (1)
**Age, years**	Mean 10.6 ± 3.0 Median 11 [9–13.25]	Mean 8.6 ± 3.9 Median 9 [7–12]

Abbreviation: CM, castrated male; F, female; M, male; SF, spayed female. Data are mean ± SD; median for age.

**Table 2 animals-15-02004-t002:** Complete blood count and serum biochemistry in dogs enrolled in this study.

Parameter	Reference Range	Diabetic Mellitus (n = 7)	Control (n = 4)
**WBC**	5.05–16.76 (k/µL)	10.53 (6.70–13.65)	10.01 (8.65–14.86)
**RBC**	5.65–8.87 (M/µL)	7.31 (6.48–8.32)	7.24 (6.04–8.33)
**HCT**	37.3–61.7 (%)	45.00 (43.90–56.00)	46.15 (40.10–54.68)
**Hb**	13.1–20.5 (g/dL)	15.00 (14.20–18.60)	15.60 (13.60–18.05)
**MCH**	21.2–25.9 (pg)	22.50 (20.50–23.30)	22.15 (21.15–22.55)
**MCHC**	32.0–37.9 (g/dL)	33.30 (32.70–33.40)	33.85 (33.05–33.90)
**PLT**	148–484 (k/µL)	411.00 (369.00–681.00)	357.00 (264.80–535.50)
**TP**	5.0–7.2 (g/dL)	5.90 (5.60–6.60)	7.00 (6.45–7.40)
**ALB**	2.6–4.0 (g/dL)	3.00 (2.90–3.20)	3.20 (2.98–3.65)
**BUN**	9.2–29.2 (mg/dL)	19.60 (14.20–21.80)	24.30 (19.93–30.85)
**CREA**	0.4–1.4 (mg/dL)	0.43 (0.35–0.50)	0.55 (0.48–0.73)

Data are presented as the median and interquartile range. No parameter differed significantly between groups (all *p* > 0.05).

**Table 3 animals-15-02004-t003:** Summary statistics for relevant diabetic parameters.

Parameter	Diabetic Mellitus (n = 7)	Control (n = 4)
Laboratory glycemic markers		
**GA %**	29.10 (27.10–37.50) *	17.10 (14.68–17.50)
**Fructosamine, μmol/L**	303.00 (251.00–355.00) *	214.00 (197.30–232.30)
**HbA1c, %**	4.20 (3.20–5.00) *	2.10 (2.03–2.25)
**CGM metrics**		
**Mean glucose, mg/dL**	233.00 (217.00–288.00) **	111.00 (96.75–118.50)
**TIR1, %**	8.00 (2.00–11.00) **	58.50 (43.25–77.50)
**TIR2, %**	47.00 (38.00–56.00) **	81.50 (60.50–92.75)
**TAR1, %**	85.00 (77.00–96.00) **	20.50 (3.00–32.00)
**TAR2, %**	46.00 (37.00–60.00) **	0.00 (0.00–0.00)
**TBR, %**	5.00 (2.00–7.00) *	18.50 (7.25–39.50)

Data are presented as the median and interquartile range. Abbreviations: CGM, continuous glucose monitoring; HbA1c, glycated hemoglobin A1c; TAR1, time above range > 130 mg/dL; TAR2, time above range > 250 mg/dL; TBR, time below range < 90 mg/dL; TIR1, time in range 90–130 mg/dL; TIR2, time in range 90–250 mg/dL. Value on the differences between control and diabetic mellitus group (* *p* < 0.05, ** *p* < 0.01).

## Data Availability

The original contributions presented in this study are included in the article/Appendix A; further inquiries can be directed to the corresponding author.

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
