# Peer review of "Glycated Albumin and Continuous Glucose Monitoring Metrics in Dogs with Diabetes Mellitus: A Pilot Study"

_animals, 2025, doi:10.3390/ani15142004_

Round 1

Reviewer 1 Report (Previous Reviewer 2)

Comments and Suggestions for Authors

Dear authors.

Congratulations on this new version of your manuscript. The new statistical methods are indeed more adequate (compared to the previous version). I am glad you made this change.

Author Response

Comment 1: Congratulations on this new version of your manuscript. The new statistical methods are indeed more adequate (compared to the previous version). I am glad you made this change.

Response: Thank you. We are pleased the revised statistical approach meets expectations. We hope that these changes will help our research results to be published in your journal.

Reviewer 2 Report (New Reviewer)

Comments and Suggestions for Authors

The authors carried out a very interesting study highlighting monitoring tools to managing canine diabetes. However, some points listed below need to be clarified:

Introduction

L24:  endocrine disorder.

L 28:  Do not use the same words already used in the title.

L 43-44: Check the most current etiology classification of DM carried out by the ALIVE project (Agreeing Language in Veterinary Endocrinology (ALIVE): Diabetes mellitus- a modified Delphi-method-based system to create consensus disease definitions: Niessen et al., 2022)

L 80-83: I suggest that the hypothesis comes above the objective.

Methodology:

L87-88: It´s not necessary to exposed the study aim again.

L119-120: Has a new continuous monitoring device been applied? Please clarify.

L123-126: These glucose ranges are unclear, please explain. What criteria were used to determine these glucose ranges (90-130 mg/dL or 90-250 mg/dL)?

L 120-121: Add previous papers that validated the use of Caresens Air (i-SENS, 120 Korea) and Freestyle Libre 2 devices in dogs

Results

L 159-164 : I suggest presenting quantitative data such as breed, sex and age through descriptive statistics (mean and standard deviation).

L 164: Present this data with descriptive statistics

L 174: Figure 1-Correlation of glycated albumin with fructosamine and HbA1c in diabetic dogs. (is this corret?) Was it just diabetics or all patients? Since N = 30

L 210: Table 3 Format table, column 1 is not aligned.

L-235: The graphs are very small. I suggest separating into 3 distinct figures.

Discussion

Needs to be improved. Few studies have been discussed. I suggest expanding research for study in other species.The  observed correlations were not adequately explained.

L 288-296: Add literature data

Comments on the Quality of English Language

The language is adequate. Please check the subject-verb agreement and in some points the use of non-formal language.

Author Response

[Introduction]

Comment 1: L24:  endocrine disorder.

Response: Thanks for your detail comments. We modified the sentence as follows:

“Diabetes mellitus (DM) is one of the most common endocrine disorders in dogs.”

Comment 2: L 28:  Do not use the same words already used in the title.

Response: Thank you for your thoughtful comments. We have rephrased the sentence below to avoid repeating the title.

“This pilot study evaluated the correlation between GA and conventional glycemic markers and continuous glucose monitoring (CGM)-derived metrics in dogs.”

Comment 3: L 43-44: Check the most current etiology classification of DM carried out by the ALIVE project (Agreeing Language in Veterinary Endocrinology (ALIVE): Diabetes mellitus- a modified Delphi-method-based system to create consensus disease definitions: Niessen et al., 2022)

Response: Thanks for your detail comments. We updated text and added Niessen et al., 2022 citation as follow.

“According to the ALIVE consensus definitions, canine diabetes mellitus is classified etiologically as either insulin-deficient (IDDM) or insulin-resistant (IRDM), with IDDM remaining the most common form seen in practice [1].”

Comment 4: L 80-83: I suggest that the hypothesis comes above the objective.

Response: Thanks for your detail comment. We agree with what you said. Therefore, we placed the phrase about hypothesis before the phrase about objective. We hope that these corrections have been appropriate.

“We hypothesized that GA would align closely with CGM-derived measures, thereby serving as a valuable adjunct to existing biochemical markers for evaluating and optimizing glycemic control in canine diabetes. Considering the need for additional biomarkers that capture short-term glucose variability, integrating CGM metrics with GA levels may offer new insights into the day-to-day management of diabetic dogs. The objectives of this study were to identify and compare GA and Continuous glucose monitoring (CGM) metrics in diabetic dogs and control groups, and to evaluate the association between GA levels and CGM metrics in diabetic dogs”

[Methodology]

Comment 5: L87-88: It´s not necessary to exposed the study aim again.

Response: Thank you for your detailed comments. We have removed the duplicated aim statement.

Comment 6: L119-120: Has a new continuous monitoring device been applied? Please clarify.

Response: Thanks for your detail comments. In this study, two CGMS devices were used, and the following features of devices were described. We replaced the previous sentence on sensor failure and added a concise description of CareSens Air to remove any ambiguity about (i) how incomplete traces were handled and (ii) whether the device itself was novel.

“Two continuous glucose monitoring (CGM) systems originally developed for human use were employed in this study: the FreeStyle Libre 2 (Abbott; n = 1), which has previously been tested in both healthy and diabetic dogs [7,8,15], and the CareSens Air (i-SENS, Seoul, Republic of Korea; n = 10), a 15-day sensor introduced in 2023 that utilizes the same interstitial fluid–based technology, has so far only been evaluated in humans [16], and was applied to dogs for the first time in this study.”

Comment 7: L123-126: These glucose ranges are unclear, please explain. What criteria were used to determine these glucose ranges (90-130 mg/dL or 90-250 mg/dL)?

Response: Thanks for your detail comments. Because no consensus CGM thresholds have yet been established for dogs, the two intervals were chosen pragmatically:

  • 90–130 mg/dL approximates fasting normoglycaemia reported for healthy dogs.
  • 90–250 mg/dL mirrors the “100–250 mg/dL” target window commonly used in clinical practice for regulated diabetic dogs.

We added the following sentence to the Methods.

“In the absence of canine-specific CGM guidelines, we selected two practical bands. 90-130 mg/dL (fasting normoglycemia) and 90-250 mg/dL (routine clinical target range for insulin-treated dogs).”

Comment 8: L 120-121: Add previous papers that validated the use of Caresens Air (i-SENS, 120 Korea) and Freestyle Libre 2 devices in dogs

Response: Thanks for your detail comments. We have revised the reference as follows. We hope that these revisions will help our research results to be published in your journal.

“Two flash continuous glucose monitoring (CGM) systems originally developed for human use were employed in this study: the FreeStyle Libre 2 (Abbott; n = 1), which has previously been tested in both healthy and diabetic dogs [7,8,15], , and the Ca-reSens Air (i-SENS, Seoul, Republic of Korea; n = 10), a 15-day sensor introduced in 2023 that utilizes the same interstitial fluid–based technology, has so far only been evaluated in humans [16], and was applied to dogs for the first time in this study.”

[Results]

Comment 9: L 159-164 : I suggest presenting quantitative data such as breed, sex and age through descriptive statistics (mean and standard deviation).

Response: Thank you for your comments. We have revised the contents of Table 1: breed and sex distributions have been added, age is now presented as mean ± SD and median, and individual data have been moved to Supplementary Table S2.

Comment 10: L 164: Present this data with descriptive statistics

Response: Thanks for your detail comments. We have added the information you pointed out to Table 1. We hope that these changes will be appropriate.

Comment 11: L 174: Figure 1-Correlation of glycated albumin with fructosamine and HbA1c in diabetic dogs. (is this correct?) Was it just diabetics or all patients? Since N = 30

Response: Thanks for your detail comments. This was the result that progressed in all patients. Therefore, it was modified as follows.

“Correlation of glycated albumin with fructosamine and HbA1c in dogs enrolled in this study (n = 30).”

Comment 12: L 210: Table 3 Format table, column 1 is not aligned.

Response: Thanks for your detail comments. We have reworked the table format and added it to the main text: Column 1 left-aligned; category rows bold; variable rows indented 0.3 cm; numeric columns centered.

Comment 13: L-235: The graphs are very small. I suggest separating into 3 distinct figures.

Response: Thank you for your detailed comments. The original Figure 3 has been divided into three separate figures: Figure 3 (GA), Figure 4 (fructosamine), and Figure 5 (HbA1c). Corresponding captions and in-text references have been updated accordingly.

[Discussion]

Comment 14: Needs to be improved. Few studies have been discussed. I suggest expanding research for study in other species. The observed correlations were not adequately explained.

Response: Thanks for your good comments. We have added the relevant literature data and citations to support the statements. Added human (Desouza 2023) and feline (Mori 2009) GA studies plus canine GA papers. We sincerely hope that these revisions will help our research results to be published in your journal.

Comment 15: L 288-296: Add literature data

Response: We have added the relevant literature data and citations to support the statements as follow. We hope that these corrections have been appropriate.

“Although fructosamine and GA both reflect short-term glycemic status (~2–3 weeks), fructosamine may be more influenced by total serum protein changes, potentially leading to under- or overestimation of glycemia in dogs with hypoproteinemia or fluctuating protein levels [19]. By contrast, GA specifically measures albumin gly-cation as a proportion of total albumin, making it less susceptible to confounding due to altered total protein [20]. HbA1c remains valuable for evaluating longer-term (~2–3 months) glycemic trends but may not capture rapid fluctuations or short-term treat-ment adjustments. Moreover, conditions that affect red blood cell turnover or hemo-globin metabolism can invalidate HbA1c results [4]. Therefore, GA may offer advantages in clinical scenarios where serum protein or RBC-related factors hinder the reliability of other markers.”

Comment 16: The language is adequate. Please check the subject-verb agreement and in some points the use of non-formal language.

Response: Thanks for the good point. We have re-examined the English language in general. We hope that our corrections are appropriate.

Reviewer 3 Report (New Reviewer)

Comments and Suggestions for Authors

The study is original and could be very relevant for the field. According to the small nr of cases it could be considered Case report or Short communication.

 The objective of this study is to evaluate the monitoring utility of glycated albumin (GA) by comparing it with the other well established biomarkers of Diabetes mellitus (DM) in dogs, like fructosamine and HbA1c,by analyzing their correlations with glucose monitoring (CGM) metrics. Authors also assessed these markers against CGM metrics, which they consider new and useful indicators for diagnosing and managing diabetes.

The methology of the study is quite usual. Authors should provide some more details about the method of determination of glycated albumin.

The results could be very important if the study is made on a larger cohort of cases. For such a small nr of cases the statistic is irrelevant. In Abstract the authors are mentioning that a total of 30 dogs were included in this prospective pilot study, comprising insulin-dependent diabetic dogs (n = 10) and 29 healthy controls (n = 20), but in reality after owners consentjust 11 cases were included (diabetic dogs, n = 7; control dogs, n = 4).

Table 2- no significant differences between Diabetic and control group?

The conclusions should be improved after correcting the major weakness mentioned previously.

The references should be written according to MDPI Instructions for authors.

For Introduction, improvement of Methodology and Discussions authors may see also

1.Screening diabetic cats for hypersomatotropism: performance of an enzyme-linked immunosorbent assay for insulin-like growth factor 1, Journal of Feline Medicine and Surgery 2014 16: 82-88, DOI: 10.1177/1098612X13496246

Madalina Rosca, Mihai Musteata, Carmen Solcan, Gabriela Dumitrita Stanciu, Gheorghe Solcan, Feline Hypersomatotropism, an Important Cause for the Failure of Insulin Therapy, Bulletin UASVM Veterinary Medicine 71(2) / 2014, 298-304, DOI:10.15835/buasvmcn-vm: 10288

Author Response

Comment 1: The study is original and could be very relevant for the field. According to the small nr of cases it could be considered Case report or short communication. The objective of this study is to evaluate the monitoring utility of glycated albumin (GA) by comparing it with the other well-established biomarkers of Diabetes mellitus (DM) in dogs, like fructosamine and HbA1c, by analyzing their correlations with glucose monitoring (CGM) metrics. Authors also assessed these markers against CGM metrics, which they consider new and useful indicators for diagnosing and managing diabetes.

Response: We sincerely thank the reviewer for the constructive suggestion to consider a shorter article type. Although our sample size is modest, the study was prospectively designed, includes a controlled comparison, and evaluates multiple biomarkers alongside continuous glucose monitoring. Presenting these elements in a full research article allows us to describe the methodology and statistical analysis in sufficient detail for reproducibility. We therefore respectfully prefer to retain the current format, while remaining fully willing to follow the editors’ guidance should an alternative article type be considered more appropriate.

Comment 2: The methology of the study is quite usual. Authors should provide some more details about the method of determination of glycated albumin.

Response: Thanks for your detail comments. We have added further details in the Materials and Methods section regarding how glycated albumin was measured as follow. We hope that these corrections have been appropriate.

“GA levels were measured using a cartridge-based system specifically validated for use in dogs and cats. The assay required 5 μL of serum or plasma treated with lithium heparin.”

Comment 3: The results could be very important if the study is made on a larger cohort of cases. For such a small nr of cases the statistic is irrelevant. In Abstract the authors are mentioning that a total of 30 dogs were included in this prospective pilot study, comprising insulin-dependent diabetic dogs (n = 10) and 29 healthy controls (n = 20), but in reality after owners consentjust 11 cases were included (diabetic dogs, n = 7; control dogs, n = 4).

Response: Thanks for your detail comments. To avoid confusion, the Abstract now distinguishes the total study population (30 dogs: 10 diabetic, 20 control) from the subset that underwent CGM analysis (11 dogs: 7 diabetic, 4 control). We have reordered the sentences, so the overall enrollment is stated first, followed immediately by the CGM subset numbers. All corresponding sections have been updated, and the Discussion explicitly notes the limited size of the CGM subset as a study limitation.

Comment 4: Table 2- no significant differences between Diabetic and control group?

Response: Thanks for your comments. The results shown in Table 2 showed no significant difference between the diabetes and control groups. Added statistical comparisons and p-values; clarified non-significant items.

Comment 5: The conclusions should be improved after correcting the major weakness mentioned previously.

Response: Thanks for your good comments. Conclusions revised to match data and acknowledge limitations as follow. We hope that these corrections have been appropriate.

“In conclusion, despite the limitations of our pilot study—including a small sample size and a relatively homogeneous population—our findings suggest that glycated albumin (GA), alongside fructosamine and HbA1c, offers valuable insight into glycemic control over a two-week period. Notably, GA showed strong associations with both Mean Glucose and CGM-derived extremes such as TIRâ‚‚ and TARâ‚‚, supporting its potential as a complementary short-term glycemic marker in diabetic dogs. Given GA’s relative stability in conditions affecting serum protein or red blood cell turnover, it may offer clinical advantages in patient populations where traditional glycemic markers are less reliable or practical.”

Comment 6: The references should be written according to MDPI Instructions for authors.

Response: Thanks for your detail comment. All references formatted per MDPI guidelines.

Comment 7: For Introduction, improvement of Methodology and Discussions authors may see also

1.Screening diabetic cats for hypersomatotropism: performance of an enzyme-linked immunosorbent assay for insulin-like growth factor 1, Journal of Feline Medicine and Surgery 2014 16: 82-88, DOI: 10.1177/1098612X13496246

Madalina Rosca, Mihai Musteata, Carmen Solcan, Gabriela Dumitrita Stanciu, Gheorghe Solcan, Feline Hypersomatotropism, an Important Cause for the Failure of Insulin Therapy, Bulletin UASVM Veterinary Medicine 71(2) / 2014, 298-304, DOI:10.15835/buasvmcn-vm: 10288

Response:

Thank you for these valuable references. We have incorporated both papers into the Introduction and Discussion. In response to your suggestion to enhance flow and clinical relevance, we added two sentences that (i) highlight the successful use of the IGF-1 ELISA as an economical screen for feline hypersomatotropism and (ii) draw a parallel with GA’s potential as an accessible, short-term biomarker in routine canine practice. These additions emphasize the translational value of affordable diagnostics across species and better frame GA’s real-world applicability.

“In feline medicine, the use of a simple IGF-1 ELISA has tripled the detection rate of hypersomatotropism, demonstrating how an affordable serum test can significantly improve case identification [11,21]. Similarly, GA could serve as a rapid screening tool in general practice, with CGM reserved for complex or referral cases.”

Round 2

Reviewer 3 Report (New Reviewer)

Comments and Suggestions for Authors

The authors have made all the corrections suggested. I recommend the acceptance of the article in actual revised form

This manuscript is a resubmission of an earlier submission. The following is a list of the peer review reports and author responses from that submission.

Round 1

Reviewer 1 Report

Comments and Suggestions for Authors

This study investigates the use of glycated albumin (GA) as a potential biomarker for short-term glycemic control in dogs with diabetes mellitus (DM). The manuscript provides valuable insights into the correlation between GA, fructosamine, hemoglobin A1c (HbA1c), and continuous glucose monitoring (CGM) metrics. The research is well-structured, and the methodology is appropriately detailed. However, there are areas that require improvement, particularly in statistical analysis, sample size considerations, and discussion depth.

  1. Relevance and Novelty: The study addresses an important gap in veterinary diabetes management by evaluating GA as a short-term glycemic marker. The integration of CGM metrics adds a novel perspective.
  2. Methodology: The inclusion of multiple glycemic markers (GA, fructosamine, HbA1c) and CGM enhances the study's robustness. The selection of diabetic and control groups is appropriate.
  3. Findings: The significant correlations observed between GA and other glycemic markers support its potential utility in clinical practice.

Areas for improvment:

         1. Sample Size and Statistical Analysis:

The study sample is relatively small (n=30), which limits the generalizability of the findings. The authors should acknowledge this limitation more explicitly in the discussion.

A Receiver Operating Characteristic (ROC) curve analysis to determine GA cut-off values for glycemic control would strengthen the study.

  1. Discussion and Interpretation:

The discussion should include a more detailed comparison of GA with traditional markers, highlighting clinical scenarios where GA would be superior.

Consider expanding on the implications of GA variability due to albumin turnover and protein metabolism in diabetic dogs.

  1. CGM Utilization:

While CGM is well-integrated into the study, a brief discussion on its practicality and cost-effectiveness for routine veterinary practice would be beneficial.

The impact of stress-related hyperglycemia on CGM readings should be mentioned.

  1. Formatting and Clarity:

Some sections, particularly in the results, could benefit from more concise presentation.

The introduction could be streamlined to focus more on the rationale for including GA as a biomarker.

The manuscript presents a valuable contribution to veterinary diabetes research, and the findings support the role of GA in short-term glycemic monitoring. However, addressing the above concerns will significantly enhance the manuscript’s clarity, impact, and applicability. I recommend major revisions to refine the statistical analyses, expand the discussion, and improve overall presentation before acceptance for publication.

Author Response

Dear. Reviewer 1

Thank you so much for giving opportunity to revise our manuscript “Glycemic albumin and continuous glucose monitoring metrics in dogs with diabetes mellitus: a pilot study” for Animals. We want to extend our appreciation to you and the reviewers for taking the time and effort necessary to provide such insightful guidance. We have carefully considered comments offered by the reviewers. Herein, we explain how we revised the paper based on those comments and recommendations. We look forward to working further with you and the reviewers to move this manuscript closer to publication.

COMMENT 1. This study investigates the use of glycated albumin (GA) as a potential biomarker for short-term glycemic control in dogs with diabetes mellitus (DM). The manuscript provides valuable insights into the correlation between GA, fructosamine, hemoglobin A1c (HbA1c), and continuous glucose monitoring (CGM) metrics. The research is well-structured, and the methodology is appropriately detailed. However, there are areas that require improvement, particularly in statistical analysis, sample size considerations, and discussion depth.

Relevance and Novelty: The study addresses an important gap in veterinary diabetes management by evaluating GA as a short-term glycemic marker. The integration of CGM metrics adds a novel perspective.

Methodology: The inclusion of multiple glycemic markers (GA, fructosamine, HbA1c) and CGM enhances the study's robustness. The selection of diabetic and control groups is appropriate.

Findings: The significant correlations observed between GA and other glycemic markers support its potential utility in clinical practice.

RESPONSE: Thank you for recognizing the relevance and novelty of our work. We have further emphasized GA’s potential role in capturing short-term glucose fluctuations in the Discussion section. We hope that these corrections have been appropriate.

COMMENT 2. Sample Size and Statistical Analysis: The study sample is relatively small (n=30), which limits the generalizability of the findings. The authors should acknowledge this limitation more explicitly in the discussion.

RESPONSE: We fully acknowledge that the relatively small sample size (n=30) may limit the generalizability of our findings. This limitation is now explicitly addressed in the Discussion, along with recommendations for future larger-scale studies that include diverse patient populations. We have made the following changes. We hope that these changes will be appropriate.

“Second, our sample size was relatively small, and we did not include dogs with multiple comorbidities or conditions that might affect albumin turnover or red blood cell lifespan. These factors could influence GA, fructosamine, or HbA1c levels and potentially alter their correlations with CGM metrics. Including a broader population of canine patients would help validate the clinical applicability of GA and other glycemic indices in different scenarios, such as kidney disease, protein-losing nephropathy, or systemic inflammatory conditions. Therefore, further studies with larger cohorts, various comorbidities, and broader clinical contexts are required to validate the clinical applicability of GA and establish evidence-based guidelines for its integration into routine diabetes management protocols in canine practice.

COMMENT 3. Sample Size and Statistical Analysis: A Receiver Operating Characteristic (ROC) curve analysis to determine GA cut-off values for glycemic control would strengthen the study.

RESPONSE: Thanks for your detail comments. We performed a ROC analysis to determine the GA cut-off value (~23.5%), which showed excellent diagnostic accuracy (AUC = 0.97). However, we clearly noted in the Discussion that the practical clinical significance of this cut-off requires cautious interpretation due to our study’s selection criteria, which clearly distinguished diabetic dogs from controls at baseline. We discussed the need for future research in broader clinical contexts.

COMMENT 4. Discussion and Interpretation: The discussion should include a more detailed comparison of GA with traditional markers, highlighting clinical scenarios where GA would be superior.

RESPONSE: Thanks for your good comments. We expanded the Discussion by adding a dedicated paragraph comparing GA to fructosamine and HbA1c, highlighting specific clinical scenarios in which GA might be superior due to its relative stability despite changes in serum protein or red blood cell turnover.

COMMENT 5. Discussion and Interpretation: Consider expanding on the implications of GA variability due to albumin turnover and protein metabolism in diabetic dogs.

RESPONSE: Thanks for your detail comments. We also explicitly discussed how alterations in albumin turnover (e.g., protein-losing nephropathy, hepatic insufficiency) may impact GA values, emphasizing caution in interpreting GA results under these conditions. We hope these corrections are appropriate.

COMMENT 6. CGM Utilization: While CGM is well-integrated into the study, a brief discussion on its practicality and cost-effectiveness for routine veterinary practice would be beneficial.

RESPONSE: Thanks for your detail comments. In the Discussion, we note that CGM’s high cost and technical requirements limit widespread use. We then position GA measurement as a simpler and less time-intensive marker that can complement CGM or serve as an alternative when continuous monitoring is impractical. We have made the following changes. We hope that these changes will be appropriate.

“Continuous glucose monitoring systems (CGMS) allow real-time monitoring of glucose levels, enabling timely responses to glucose fluctuations [5]. These small sensors, placed beneath the skin, measure interstitial fluid glucose and help refine insulin dosages or dietary plans by generating critical alerts for significant deviations. CGMS can also aid in identifying asymptomatic hypoglycemia or hyperglycemia, ultimately enhancing insulin therapy for each dog. However, the cost and technical demands of CGMS may limit its routine application in many veterinary settings, prompting a search for additional short-term biomarkers that are simpler to measure.”

COMMNET 7. CGM Utilization: The impact of stress-related hyperglycemia on CGM readings should be mentioned.

RESPONSE: Thanks for your comments. We added the statement: “Moreover, stress-induced hyperglycemia in some dogs can temporarily inflate CGM readings, potentially misleading clinical interpretation. GA...is less affected by transient spikes in glucose.” This highlights how stress can confound CGM data.

COMMMENT 8. Formatting and Clarity: Some sections, particularly in the results, could benefit from more concise presentation.

Response: Thanks for your detail comments. We removed repetitive numeric details already presented in tables. For instance, we now summarize key findings in the text (e.g., “Mean GA was significantly higher...”) without reiterating every data point from the tables. We hope that these changes will help our research results to be published in your journal.

COMMENT 9. Formatting and Clarity: The introduction could be streamlined to focus more on the rationale for including GA as a biomarker.

RESPONSE: Thanks for your detail comments. The Introduction still provides background on canine diabetes but condenses overly detailed explanations, pivoting quickly to why GA is needed and how it differs from fructosamine/HbA1c. We hope that these changes will help our research results to be published in your journal.

Again, we appreciate all of your insightful comments. We worked hard to respond to them. Thank you for taking the time and energy to help us improve this manuscript.

Reviewer 2 Report

Comments and Suggestions for Authors

Dear authors, I have carefully revised your manuscript “Glycemic albumin and continuous glucose monitoring metrics in dogs with diabetes mellitus: a pilot study”.

I have one major problem with the statistics in this study and thus, I have recommended a major revision. I did not complete my revision (final of the discussion and conclusions) because I really think that this statistical flaw should be corrected (and it is quite possible that some/many conclusions would change if corrected).

My main concern is this.- Were all of the data normally distributed?

I can see that the authors have only used mean and standard deviation in every table (which means that all of the parameters are normally distributed). I can also see that every statistical test used is indicated for normally distributed data.

However, this is really really strange for many hematological and biochemical parameters.

Moreover (this is the most important part), this is almost totally impossible in groups as little as 4 animals. When you are comparing data in your animals with continuous control monitoring, you are comparing 4 animals vs 7 animals. Again, a normal distribution of hematological/biochemical parameters in a group of 4 animals is almost never found.

Note that when we compared 2 groups (e.g. diabetic WBC vs control WBC) we have to determine that both groups are normally distributed (not the WBC of the entire population).

Please ensure that these parameters (in these really small groups) are normally distributed (if they are not, they should be presented using median values and interquartile ranges, and statistics should also be different, different tests should be used to compared groups and to calculate correlations). If a non-normal group has to be compared to a normal one, transformation of data can be implemented.

Again, this is quite important and could (very possible) affect your results (hence your entire discussion and conclusions).

Author Response

Dear. Reviewer 2

Thank you so much for giving opportunity to revise our manuscript “Glycemic albumin and continuous glucose monitoring metrics in dogs with diabetes mellitus: a pilot study” for Animals. We want to extend our appreciation to you and the reviewers for taking the time and effort necessary to provide such insightful guidance. We have carefully considered comments offered by the reviewers. Herein, we explain how we revised the paper based on those comments and recommendations. We look forward to working further with you and the reviewers to move this manuscript closer to publication.

COMMENT 1. I have one major problem with the statistics in this study and thus, I have recommended a major revision. I did not complete my revision (final of the discussion and conclusions) because I really think that this statistical flaw should be corrected (and it is quite possible that some/many conclusions would change if corrected). My main concern is this.- Were all of the data normally distributed? I can see that the authors have only used mean and standard deviation in every table (which means that all of the parameters are normally distributed). I can also see that every statistical test used is indicated for normally distributed data. However, this is really really strange for many hematological and biochemical parameters. Moreover (this is the most important part), this is almost totally impossible in groups as little as 4 animals. When you are comparing data in your animals with continuous control monitoring, you are comparing 4 animals vs 7 animals. Again, a normal distribution of hematological/biochemical parameters in a group of 4 animals is almost never found. Note that when we compared 2 groups (e.g. diabetic WBC vs control WBC) we have to determine that both groups are normally distributed (not the WBC of the entire population). Please ensure that these parameters (in these really small groups) are normally distributed (if they are not, they should be presented using median values and interquartile ranges, and statistics should also be different, different tests should be used to compared groups and to calculate correlations). If a non-normal group has to be compared to a normal one, transformation of data can be implemented.

RESPONSE: Thank you for this important observation. We have thoroughly reassessed normality assumptions for all parameters within individual groups using the Shapiro–Wilk test. Non-parametric methods (Mann–Whitney U test) were applied to variables that violated normality (fructosamine and HbA1c levels in the control group). Additionally, we revised all relevant data presentations to use median and interquartile ranges for these parameters, reflecting more appropriate statistical methodologies given our small group sizes. The Results section and statistical methods description have been revised accordingly, clearly specifying the application of non-parametric tests where necessary.

We agree that ensuring the correct application of statistical methods is crucial for accurate interpretation and appreciate your guidance in improving the robustness of our analyses.

Sincerely yours,

Round 2

Reviewer 1 Report

Comments and Suggestions for Authors

Dear Authors,

Thank you for your detailed responses and the revisions made to the manuscript. I appreciate the effort you have put into addressing the comments and improving the quality of the work.

After reviewing the revised version, I find that the manuscript has been significantly improved and meets the necessary standards. In my opinion, the current version is suitable for acceptance.

Reviewer 2 Report

Comments and Suggestions for Authors

Dear authors. I will explain my problem with statistics in a different way (because I think that I have not been understood).

  • Parametrically distributed data should be expressed as mean and SD (ideally SEM).
  • Non-parametrically ones should be expressed as median and interquartile.
  • When comparing 2 datasets, if both are normally distributed, we would used certain tests; if both are non-parametrical we would use DIFFERENT tests. If we are comparing a normal dataset with a non-normal one, we have to transform one of them.

Reading your manuscript, you have displayed all of the data (for example, Table 2) as NON-parametric (before my suggestion all of them were presented as normal). This is different to your comments in 2.5. Moreover, in this section 2.5, the authors explain that only fructosamine and Hba1c in the control group are not-normal). Thus, you cannot compare these data with fructosamine or Hba1c of the other groups (which are normal), you should transform them.

I understand that the statistics here is difficult, but this type of problem is quite common when the number of animals is so small.

Using wrong statistical tests would led you to wrong conclusions, hyping false results and hiding real ones.

PS.- Correlations should also be between groups with the same type of distribution (cannot mix a non-normal dataset with a normal one).